# Users' Experiences with the Use of Transaction Data to Estimate Consumption-Based Emissions in a Carbon Calculator

**Wolmet Barendregt** [1,*,†] , **Aksel Biørn-Hansen** [2,†] **and David Andersson** [3,†]

1. Department of Industrial Engineering and Innovation Sciences, Eindhoven University of Technology, 5600 MB Eindhoven, The Netherlands
2. Department of Human Centered Technology, KTH Royal Institute of Technology, 100 44 Stockholm, Sweden; aksbio@kth.se
3. Department of Psychology, University of Gothenburg, 413 14 Gothenburg, Sweden; david.s.andersson@psy.gu.se
* Correspondence: w.barendregt@tue.nl; Tel.: +31-6-271-55954
† All authors contributed equally to this work.

**Abstract:** With global greenhouse gas (GHG) emissions ever increasing, we are currently seeing a renewed interest in carbon footprint calculators (or carbon calculators for short). While carbon calculators have traditionally calculated emissions based on user input about e.g., food, heating, and traveling, a new development in this area is the use of transaction data to also estimate emissions based on consumption. Such carbon calculators should be able to provide users with more accurate estimations, easier input possibilities, and an incentive to continue using them. In this paper, we present the results from a survey sent to the users of such a novel carbon calculator, called Svalna. Svalna offers users the possibility to connect their bank account. The transaction data are then coupled with Environmental Extended Multi Regional Input Output data (EE-MRIO) for Swedish conditions which are used to determine a continuous overview of the user's greenhouse gas emissions from consumption. The aim of the survey was to (a) understand whether people are willing to connect their bank account, (b) whether they trust the calculations of their emissions, and (c) whether they think the use of Svalna has an effect on their behaviour. Furthermore, we wanted to know how Svalna could be improved. While the results of the survey showed that many users were willing to connect their bank account, a rather large part of the users perceived safety risks in doing so. The users also showed an only average level of trust in the correctness of the estimated greenhouse gas emissions. A lack of trust was attributed to experiencing technical problems but also to not knowing how the emissions were calculated and because the calculator could not capture all details of the user's life. However, many users still indicated that the use of Svalna had helped them to initiate action to reduce their emissions. In order to improve Svalna, the users wanted to be able to provide more details, e.g., by scanning receipts and get better options for dealing with a shared economy. We conclude this paper by discussing some opportunities and challenges for the use of transaction data in carbon footprint calculators.

**Keywords:** carbon footprint calculator; transaction data; consumption; trust

## 1. Introduction

Climate change is arguably the most pressing societal challenge of our time and poses a serious threat to ecosystems and the future well-being of humanity [1]. Despite the urgency to mitigate our environmental footprint, global greenhouse gas (GHG) emissions are still increasing [2] and

a recent analysis also projects them to increase in the future [3]. The failure to address this issue through establishing internationally binding abatement targets has led the international community to reconsider the top-down approach and instead build the Paris agreement on nationally determined contributions (NDCs). This puts the need for public engagement and support in national policy processes at the center stage. New climate movements are emerging in different parts of the world. The Fridays For Future movement [4], for example, calls on global leaders to take responsibility, but also underlines the need for personal emission abatement from all. Indeed, the goal to reduce our personal carbon footprint from 10 tonnes $CO_2$eq per capita and year in Sweden (and 28 in the U.S.) to 1 tonne $CO_2$eq/cap/yr by 2050 to prevent global warming from going over 2 °C can probably only be reached when working on both fronts. While carbon footprint calculators (or carbon calculators for short) to measure individual GHG emissions have been available for more than a decade, the renewed focus on public engagement has given rise to a new interest in the use of individual carbon footprint calculators. Carbon footprint calculators are tools that seek to measure and give feedback on the GHG emissions of activities and/or the lifestyle choices of individuals, with the aim of promoting more pro-environmental behaviour in people. It is common to harness techniques from behaviour psychology in order to do so, such as information provision, prompts, social comparisons and commitments. These techniques tap into several motivational drivers thought to influence behaviour.

There are many carbon calculators publicly available today, developed for research, business or non-profit purposes [5,6]. Notable examples that target a global audience include the Footprint calculator of the Global Footprint Network [7], World Wildlife Fund's climate calculator [8], the CoolClimate calculator of the University of California [9] and the carbon footprint calculator of the United Nations [10]. In addition, there are several carbon footprint calculators that have been developed for the living conditions in specific countries. Scandinavia hosts several different examples of carbon footprint calculators, including Ducky in Norway [11], Tomorrow in Denmark [12], Sitra's Finnish lifestyle test [13], and Deedster in Sweden [14].

While these carbon footprint calculators primarily aim to increase the user's knowledge about their emissions, other calculators, such as those of the MyClimate foundation [15] and ClimateCare [16], primarily aim to help users offset their emissions by investing in carbon compensation projects.

Most carbon footprint calculators aim to estimate a user's GHG emissions based on questions about e.g., traveling, heating and eating patterns (e.g., vegan, vegetarian, omnivore). However, consumption of goods, such as clothes and mobile phones, also plays an important role in the size of an individual's emissions, especially in wealthy countries [17]. This is a factor that most carbon footprint calculators do not take into account, and some carbon calculators, such as the calculator by Milieu Centraal [18], therefore explicitly acknowledge that their results do not include emissions based on the purchase of goods.

However, there is currently a rise in the development of carbon footprint calculators that try to offer insight into emissions caused by consumption. These calculators make use of e.g., transaction data from bank statements to offer individuals a detailed insight into their emissions based on consumption. For example, Nordea, a European financial services group, has included a carbon calculator in one of their platforms that calculates the approximate $CO_2$ impact stemming from the goods and services people have bought with their credit and debit cards [19]. While this calculator is not based on individual products as data, another initiative is ICA's My Climate Goal tool [20]. This tool was launched in 2018, and allows shoppers of ICA supermarkets to see their $CO_2$ emissions based on food purchases. Several newly released or to be released calculators, such as My Carbon Action in Finland [21], and Joro in the U.S. [22] and Doconomy in Sweden [23] are other examples of this trend.

However, since the use of more detailed and continuous data in carbon calculators is a recent trend, research into the use of this new type of calculators is still fairly limited. Svalna (meaning "to cool down", in Swedish), is a similar kind of calculator, which aims to to continuously provide reliable estimates of users' GHG emissions, including those related to consumption, over time by employing

transaction data from the users' bank statements, together with registry data and data inputted by the users themselves. However, unlike the calculators mentioned above, Svalna has already been operational for a while and has currently over 17,000 registered users, of which 4000 on their recently launched mobile application. In this paper, we want to investigate users' experiences with the use of Svalna as an example of such a new kind of calculator. Specifically, we aim to address the following research questions:

- Are people willing to connect their bank account in order to receive a more detailed calculation of their GHG emissions based on transaction data?
- Do people trust the calculations of their GHG emissions?
- Do people think that using a carbon calculator like Svalna has an effect on their behavior?
- How could Svalna be improved?

While this paper addresses these questions by investigating people's reactions to Svalna as an example of the new type of carbon calculators, we aim to highlight the opportunities and challenges of calculating people's carbon footprint using transaction data in general. This knowledge may be valuable for other research and development of this new kind of carbon calculators.

## 2. Background

### 2.1. What Is a Carbon Footprint Calculator?

In this section, we will shortly discuss the design of carbon footprint calculators and some previous work on the effectiveness of carbon calculators. However, before we do so, we have to address the fact that we have used the terms 'carbon calculator' or 'carbon footprint calculator' without much clarification. Carbon footprint or carbon calculators are a form of environmental footprint calculators. Such environmental footprint calculators estimate their users' environmental footprint based on information about their lifestyle (e.g., housing, expenses, travel habits, diet). This environmental footprint is a measure of environmental impact, which includes greenhouse gas emissions, as well as water and land use. The emission of greenhouse gases, such as carbon dioxide ($CO_2$), nitrous oxide, methane, and others, is the cause for the increase of global temperatures. As the name suggests, carbon calculators or carbon footprint calculators focus on the emission of ($CO_2$) as part of the users' environmental footprint. However, some carbon footprint calculators do indeed take other greenhouse gases into account as well, although this is not always communicated explicitly to the user [5]. Although we are aware of the slightly misleading term, we will write 'carbon footprint calculator' and 'carbon calculator' interchangeably to indicate a tool that calculates greenhouse gas emissions in the broadest sense.

### 2.2. The Design and Use of Carbon Footprint Calculators

In a review of 31 internet-based carbon calculators, Bottrill [24] highlighted the diversity of existing tools, with different calculators displaying a varying degree of complexity and ways in which the user can interact with the tool, ranging from filling in a simple spreadsheet to web interfaces with a multi-step process. The type of results that carbon calculators provide to the user is on the other hand fairly homogeneous, with most mainly reporting the annual impact on the environment by the user [25]. Beyond providing an overview and breakdown of the annual GHG emissions of a person, comparisons to national averages, tips and pledges, as well as options for carbon offsetting are common features in newer carbon calculators (e.g., [6]). This type of features often relates back to and makes use of intervention techniques from behavioural psychology, such as feedback, information provision, prompts, social comparison and goal setting [26].

There are only few studies on the effectiveness of using carbon calculators, and their results are mixed. Some studies have reported actual reductions amongst their participants, e.g., [27,28]. However, others have questioned whether carbon calculators really have an effect on people's

behaviour. A longitudinal study by Büchs et al. [29] where participants were introduced to a carbon calculator during an interview, significantly increased awareness. However, it did not result in any measurable reductions in energy use.

As described in the introduction, Svalna works in a different way than the carbon calculators involved in the research presented above. It aims to provide users with a detailed insight into their GHG emissions based on their consumption by making use of transaction data from bank statements. The design of Svalna will be presented in more detail in the next subsection.

## 2.3. The Design of Svalna

Svalna is created by Svalna Inc., a research-based company based in Gothenburg, Sweden. Svalna aims to simplify and automate all aspects of data collection to estimate the user's GHG emissions without compromising the quality of the data. The service relies on data from four primary sources in order to estimate a user's GHG emissions:

1. financial transaction data from the user's bank paired with data on GHG emissions per monetary unit for a number of categories estimated using Environmental Extended Multi Regional Input Output analysis (EE-MRIO) for Swedish conditions. EE-MRIO analysis is used to provide sector specific estimations of total greenhouse gas emissions from the material inputs required along the entire supply chain, including both direct and indirect flows.
2. data on fuel type, fuel consumption and yearly mileage from the Swedish Transport Agency via the registry number of the users car,
3. data about the average energy performance of the user's apartment building from the National Board of Housing, Building and Planning via the user's home address
4. data entered by the user, covering things such as dietary choices, food waste, physical activity (which affects model assumptions about food intake), commuting habits and so on.

By allowing users to connect their private bank accounts and credit cards to the tool, Svalna can estimate GHG emissions of expenditures within different consumption categories. All data communication is encrypted, and data are stored on secure servers in accordance with Swedish and EU regulations (General Data Protection Regulation, GDPR). Data collection from banks is enabled via open banking platform Tink [30] and Swedish electronic-ID provider "Bank ID" [31]; an electronic identification solution that allows individuals to authenticate and conclude agreements with, e.g., companies and governments agencies, over the Internet. Similar citizen identification solutions exist in most European countries and the Payment Service Directive 2 of the European Union (2015/2366) [32] requires that all financial institutions operating within the EU facilitate for their customers to be able to share data with third parties such as Svalna. Svalna's technical solution is hence scaleable to other countries in the EU. Financial transaction data provide a detailed account of a user's expenditures. All transactions are classified according to a modified version of the Classification of Individual Consumption According to Purpose (COICOP) scheme developed by the UN statistics division [33]. Whenever a transaction cannot be classified automatically, the users are asked to classify the transaction themselves. User-classified transactions then add intelligence to the system's classification algorithm, in the sense that future purchases by other users can be better pre-classified. Users can also re-classify or remove transactions that are not considered relevant. Transaction data are available for between three months and five years back in time with an average of 14 months, depending on the user's bank. This means users can see how their GHG have changed historically already from the first login. For research purposes, this also provides a baseline estimate of the user's GHG emissions before they started using Svalna. Bank data can be updated every time the user logs in, and in the near future the user will be able to authorize Svalna to fetch the bank data in the background for an extended period so that emission results can be updated automatically.

The GHG emissions calculated by Svalna are divided into four main categories: (1) consumption of goods and services (purchases in non food stores), (2) transportation, (3) residential energy, and (4) food

and beverages (purchases in food stores). A detailed account of how Svalna determines GHG emissions for each transaction within each category is beyond the scope of this paper, but can be found in [34].

After entering the data, the app provides users with an overview and breakdown of their emissions in different charts, allows them to set a goal and to see how different changes to their lifestyle would impact their emissions, as well as create and be part of groups (see Figure 1). These features are related to different strategies to promote pro-environmental behaviour, including giving *feedback*, enabling the user to make a *commitment*, *goal setting* and *comparison*.

Svalna's solution largely relies on transaction data and registry data to estimate GHG emissions, but is this really more reliable? There is, as far as we know, no established standard regarding the calculation of personal carbon footprints [35] which makes it hard to evaluate and compare different carbon calculators. However, Birnik [5] has developed a set of assessment criteria for online carbon calculators. These criteria were used by Andersson [34] to evaluate Svalna where he concluded that Svalna performs "over and beyond" these criteria. It seems that solutions using transaction data have the potential to capture a significantly larger share of users' spendings and hence GHG emissions in a more correct and controllable way. By relying on "external" data sources instead of the user's recollections, this approach also avoids the risk of response biases that are likely to affect users who input data themselves. Until standards evolve and systematic comparisons between a set of carbon calculators is conducted, this remains an open issue.

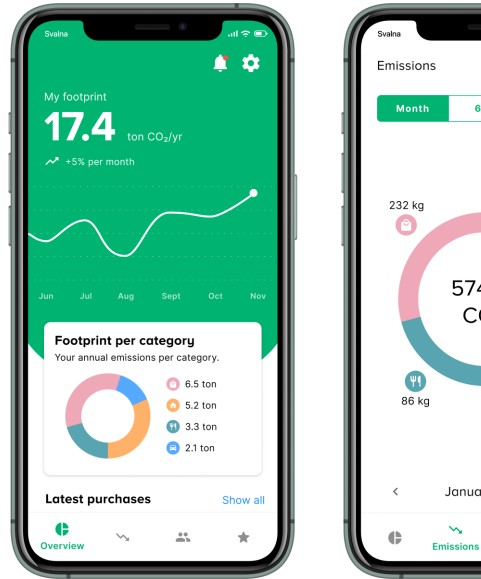 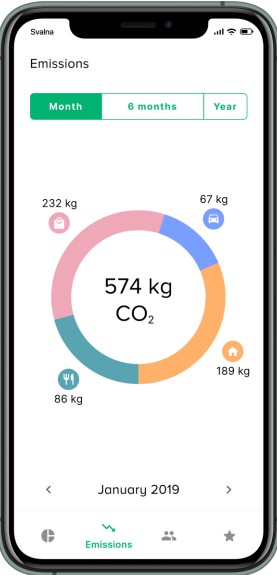 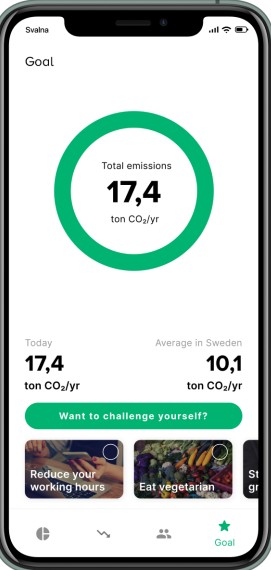

**Figure 1.** Screenshots of three features of Svalna. On the left, an overview of the user's GHG emissions, in the middle, a drill-down of the emissions in different categories, and on the right, suggestions and goal setting for reducing emissions.

## 3. Methodology

To answer the research questions presented previously, several questions were added to an online questionnaire originally created by Svalna. The whole questionnaire contained 26 questions, both open and closed. The closed questions either used a bipolar Likert-scale with five points (e.g., "No, not at all" to "Yes, definitely!"), or allowed the user to indicate several answers using multiple choice questions. Likert-scale questions asked the user to rate several statements, either separately or in a matrix with several related statements. The questions dealt with the participants' actual use of Svalna, their user experience, a self-assessment of the effect Svalna had on them, and the kind of service they would like Svalna to provide to them. The participants were also allowed to provide input about further development of Svalna.

*3.1. Participants*

The questionnaire was sent to roughly 2100 active users of Svalna on 11 September 2019. Active users in this regard were defined as users who had logged in or created a new account during the last three months up until the survey. At that time, Svalna had been released since April 2019. After the first release, several improvements and bug fixes were implemented, with the newest release in August 2019. The questionnaire was open for anonymous responses between 11 and 20 of September 2019 and received 163 answers, which gives a response rate of 7.7%. In addition, 124 out of 163 participants answered the question about their gender, with 61% of them being women, 34% men, 1% other, and 4% unwilling to reveal their gender. The majority of the participants (36%) was between 25 and 34 years old, but there were also quite some older and younger participants as well (14% 18–24, 21% 35–44, 21% 45–54, 10% 55–64, and 9% 65 years and older). Most of the participants (65%) had a job. Of the 163 responses, 126 were complete, whereas for 37 responses, one or more questions remained unanswered. In this article, we will report on both the complete and incomplete responses.

The characteristics of our sample were similar to the group of all active users at the time, where 33% were women, 28% men, and 39% unknown. Concerning age, only the age of 55% of the users was known. The majority (22%) was between 25 and 34 years old, with smaller groups among the other age intervals (11% 18–24, 13% 35–44, 9% 45–54, 4% 55–64, 2% 65+).

*3.2. Analysis*

The answers to the closed questions in the questionnaire were used as the basis for descriptive quantitative analyses using the Statistical Package for the Social Sciences (SPSS), whereas the answers from the open questions were analysed qualitatively through the creation of Affinity Diagrams [36]. This meant that all open answers were read, and relevant observations were noted down on post-it notes. These post-it notes were then grouped together in clusters in order to identify themes in this data These themes were then reviewed again to determine commonalities and larger themes.

## 4. Results

*4.1. Use of Svalna*

In order to understand how much experience the respondents had with Svalna when answering our questions, we first asked the respondents to indicate how often they had used the app during the past month. Of the 143 people that answered this question, most of them answered that they had used Svalna once or twice (Figure 2). However, another, almost equally large group, had not used Svalna during the past month.

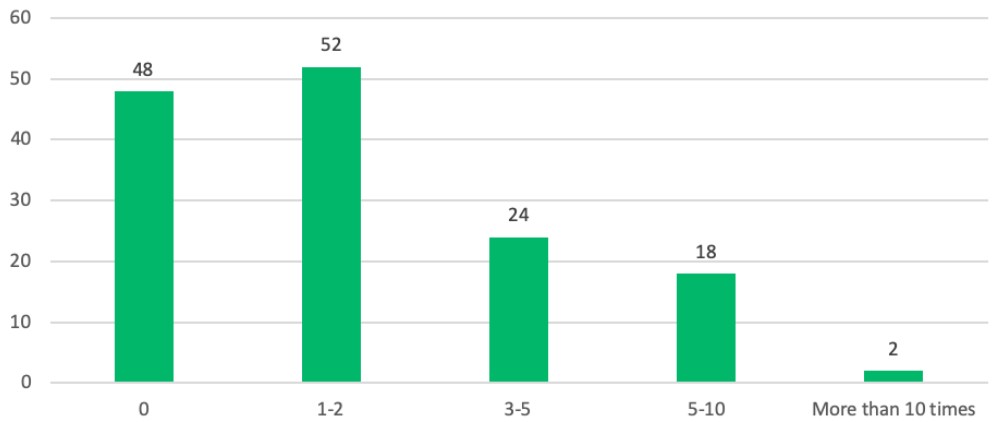

**Figure 2.** Frequency of use during the past month (*n* = 143).

### 4.2. Willingness to Connect Bank Account

When asked if the respondents had connected their bank account, most of them (70%) answered that they had indeed done so (*n* = 139). For those who had not done so, we asked for their reasons to refrain from it. Two main themes appeared in the answers from the remaining 30%. The first reason was that the participants did not think it was safe enough and expressed an uncertainty towards connecting their bank to Svalna due to a lack of trust. In addition, one respondent indicated that it was unclear why connecting a bank account would help to calculate their carbon footprint: *"[I] Would like to know more about how it works. The bank can surely not see anything else than the sums I have paid. How can one calculate CO$_2$ based on that?"* The other main reason for not connecting a bank account was that the participants experienced technical problems.

### 4.3. Perceived Correctness of Calculations

Since the intention was that the use of transaction data in combination with official registry data from government agencies and data input by the user would create a reliable picture of each individual's GHG emissions, we asked the respondents to indicate whether they thought the calculation they received was correct. The answers to this question are depicted in Figure 3.

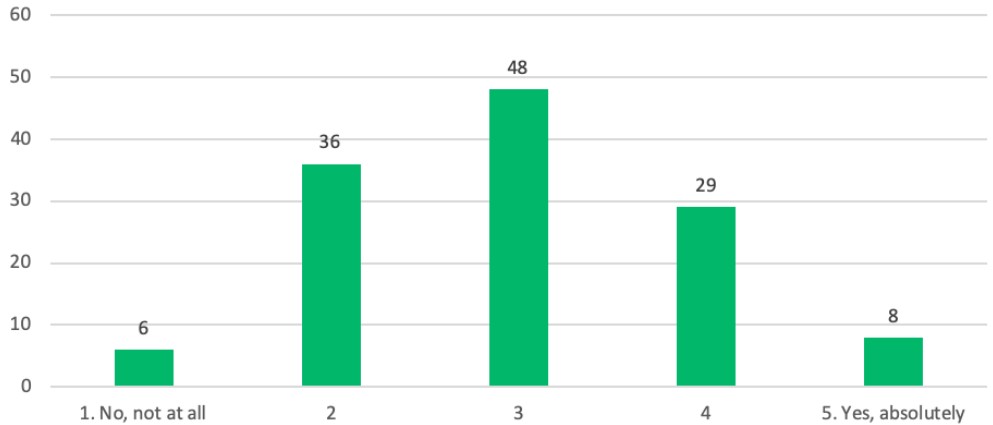

**Figure 3.** Participants' trust in the correctness of their GHG emissions (*n* = 127).

The majority of the participants did not have either a high or low trust in the correctness of their GHG emissions (Mean (M) = 2.98, Standard Deviation (SD) = 0.98, *n* = 127). To delve a bit deeper into the reasons for not trusting the calculations, the respondents who had answered that they did not think that their GHG emissions were calculated correctly (either level 1 or 2 on the scale) received a follow-up question where they could indicate why they doubted the calculation. The main reason highlighted by respondents was that they felt that the calculation did not (exactly) reflect their lives. They usually wanted the app to pay closer attention to their personal circumstances, such as sharing a household (and possibly bank account) with other people. For instance, participants highlighted that it was not possible to specify how the economic burden, and corresponding emissions, were divided between different members of a household, or to split transactions for shared costs with other people. How well the questions in the climate profile mapped onto a person's life was also mentioned, with one person saying *"...I do not think these questions give an answer to how my impact on the environment looks like overall. My impact on the environment probably looks quite different depending on what time of year it is, how my life looks like privately and professionally, etc. ..."*

In relation to the issues mentioned above, another reason for why some respondents doubted the calculations was an inaccurate categorisation of transactions in the app. Incorrectly categorised transactions sometimes led to higher emissions and impacted how reliable some participants judged the calculations. One participant put it like this: *"We moved in April and everything connected to the*

*purchase of the house resulted in enormous emissions in the app. It is obvious that you emit when moving, but I would want to have more possibilities to exclude some expenses from [my] bank account in order to get a reasonable picture of my everyday emissions during the year".*

Understanding how Svalna calculates GHG emissions was also important for participants and impacted their trust in the calculations provided by Svalna. In particular, a lack of understanding for how their GHG emissions were calculated and what the numbers were based upon created doubt amongst some participants. For instance, one participant thought it was misleading to calculate GHG emissions per Swedish krona, since organic food and more environmentally sound products are often more expensive. Another participant wondered how specific or general the calculations were, i.e., if the calculations reflected in detail what was bought in a purchase or if it was rather based on calculation templates for specific categories. Overall, participants expressed a demand for more explanations and information about how the GHG emissions in the app were calculated.

### 4.4. Subjective Effectiveness of Svalna

We asked all respondents about their subjective assessment of Svalna's potential to help them reduce their carbon footprint. To assess this, we asked them to indicate how much they agreed with several statements about the effects of using Svalna. The results are given in Figure 4. Interestingly, the statement that participants agreed most with was related to behaviour change ('I have started to do different things to reduce my emissions').

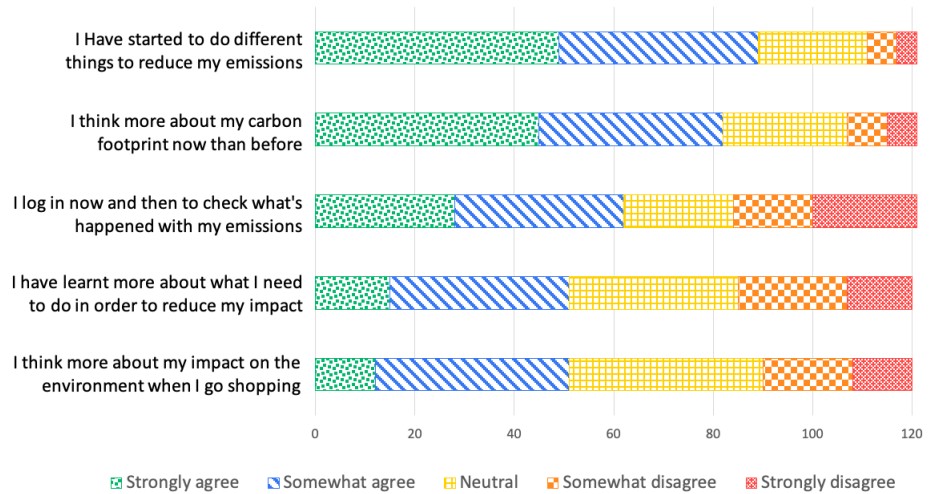

**Figure 4.** Agreement with different statements about Svalna's effectiveness. (*n* = 120).

### 4.5. Improvement of Svalna

Finally, we asked the participants what Svalna ought to focus their attention on in the future development of the service. The top three favoured developments related to improving the carbon footprint estimations in different ways. About 75% of the respondents stated that they thought "allowing for users to provide additional information in order to improve the quality of our estimation" should be a high priority (18% neutral and 7% skeptical). Related to this, 57% of the users stated that they would be interested in scanning their receipts in order to improve estimations (27% neutral and 16% skeptical). The need to handle a shared economy in the households in a smoother way was also popular, with 61% stating that this was a good idea (30% neutral and 9% skeptical).

### 5. Discussion

In this paper, we have presented the results from a questionnaire answered by 163 users of a novel carbon calculator, called Svalna. The novelty of this carbon calculator lies in the fact that it offers

users the possibility to receive a comparatively precise and continuous overview of their greenhouse gas emissions through the use of transaction data. However, the EE-MRIO data implicitly assume homogeneity with respect to price and product/service, while in reality the economy provides a variety of goods and services where the price is not always a good proxy for emissions [37]. Nevertheless, the use of transaction data can potentially lead to a more personalised and therefore more correct estimation of people's GHG emissions, and also allows for continuous estimates. This opens up for many potentially interesting techniques of engagement relating to moral nudges, norm activation, goal setting and so on. In this paper, we specifically aimed to gain a deeper understanding of people's reactions to the use of continuous transaction data using the following research questions:

- Are people willing to connect their bank account in order to receive a more detailed calculation of their GHG emissions based on transaction data?
- Do people trust the calculations of their GHG emissions?
- Do people think that using a carbon calculator like Svalna has an effect on their behavior?
- How could Svalna be improved?

In what follows, we will discuss the answers to our research questions, present some limitations and opportunities for further research, and provide some advice for the design of other carbon calculators aiming to use transaction data.

*Research Questions*

Our study showed that users were generally willing to connect their bank account. The majority of the people that did not connect their bank account was concerned about safety issues. However, it is possible that this concern was also fed by a lack of insight in the benefits of connecting a bank account, as was expressed by one of the participants. It may therefore be beneficial to provide users with some more information about the safety of the connection, as explained in the background. Furthermore, it would probably be beneficial if the user would be able to see the difference between the estimations with and without the use of transaction data.

Although the use of transaction data are meant to improve the correctness of the estimated GHG emissions, the trust in the correctness of the estimations made by Svalna was only moderate. However, we did not compare people's trust in the correctness of Svalna's estimates with their trust in the correctness of the outcomes of other carbon calculators. It is possible that people generally mistrust the outcomes of carbon calculators. The rather low level of trust should nevertheless be seen as problematic as it is likely to also affect behavioural change.

Three main reasons were given for why the participants mistrusted the results given by Svalna: technical issues, a lack of information on how the system works, and Svalna's inability to reflect the participants' lives completely. Svalna indeed experienced some technical issues with the connection to banks and displayed some obvious mis-categorisations of certain transactions at the time of the survey, which could have impacted the trust in the estimates provided by Svalna. Of course, the use of a bank connection to fetch transaction data makes Svalna more prone to technical problems than other carbon calculators that only rely on user input.

However, in many cases, the respondents indicated that their lack of trust was related to a lack of information on how emissions are calculated and that they required more information. This is a common problem in carbon calculators [5], but it remains unclear whether increased transparency of the overall system and its inner workings would strengthen the users' trust in the estimations. However, it highlights a need to understand why the estimates turn out the way they do.

Another possible explanation could be that Svalna's users are likely to perceive themselves as being "environmentally friendly" in different ways, and when faced with "bad news" in the sense that the results shown in Svalna might differ from their own assessment and self-identity, the user may experience a cognitive dissonance, which may lead them to question the estimates. Indeed, Svalna shared information that several users have commented that their emissions in Svalna

are higher than in other carbon calculators. The user might also include certain behaviours when making a self-assessment, such as choosing organic food, recycling or eco-driving. Since these behaviours only have a small effect on emissions, they are currently not specifically covered in Svalna. However, with this in mind, dealing with users' current knowledge and expectations and providing them with information about the system seems to be a necessary condition for the results of a carbon calculator to be trusted. This challenge of providing a trustworthy and representative image of an individual's GHG emissions may therefore have consequences for the use of a carbon calculator and how it affects people's behaviour. We therefore intend to further explore the relationship between people's self-assessment of their environmental friendliness with their reactions to the outcomes of a carbon calculator.

A related reason for mistrusting the calculations was that the participants did not think the GHG emissions provided by Svalna reflected their lifestyle properly, and many wanted ways to make it more fine-grained and personalised. Not surprisingly, the participants considered it a top priority for improving Svalna to allow users to provide additional information in order to improve the quality of the estimations. However, even though greenhouse gas emission estimates for specific purchases provide a more detailed and personal view of an individual's greenhouse gas emissions compared to that of earlier approaches, it is still not possible to capture all of the nooks and quirks of a person's life. An inability to customise the calculator to better reflect a person's life could lead to a user denouncing the results as false and/or stop the user from using a service. While the accuracy of calculating emissions using transaction data could be improved, and more options could be added to enable the user to further personalise the results, there is a risk that doing so will not solve the problem but instead trigger further scrutiny of the estimations. One possible way to mitigate the above mentioned challenges and concerns with using transaction data are to rather focus on social aspects or functionality that motivates users, while making use of transaction data in the background.

Finally, although the users were not convinced about the correctness of the estimations and asked for further improvements, many of them did indicate that they had started to do different things to reduce their emissions. Needless to say, the interesting question is if such a statement is in line with actual behaviours, which is not captured by this survey. However, this is something that the continuous transaction data collection should be able to verify or falsify in the future. Unfortunately, as both the first question about actual use, as well as the statement about checking your emissions regularly indicated, most users were not very active. While Svalna aims to offer a continuous overview of a user's emissions, it is necessary that users interact with it on a regular basis to see any changes. In order to make this happen, mechanisms to draw the users back to the app should be considered. The For Good app [38], for example, uses push notifications that present the user with interesting news and tips as well as calls to answer questions about the user's behaviour during the week. However, the usefulness of such an approach has not been investigated yet.

## 6. Conclusions

While the results of our survey have shown that many users were willing to connect their bank account, a rather large part of the users perceived safety risks in doing so. The users also showed an only average level of trust in the correctness of the estimated greenhouse gas emissions. A lack of trust was attributed to experiencing technical problems but also to not knowing how the emissions were calculated and because the calculator could not capture all details of the user's life. However, many users still indicated that the use of Svalna had helped them to initiate action to reduce their emissions. In order to improve Svalna, the users wanted to be able to provide more details, e.g., by scanning receipts and get better options for dealing with a shared economy.

Based on those results, we would like to offer some recommendations for others wishing to develop carbon footprint calculators that use transaction data to estimate GHG emissions based on consumption.

### 6.1. Recommendations

First of all, we would suggest to explicitly address what the user can gain by allowing the calculator to access their transaction data. This also involves explaining how the data are used to estimate the GHG emissions of every purchase as this may not be clear to the user. In addition, it could be beneficial to show the user some examples of the difference between the calculations with and without transaction data.

Second, since carbon calculators using financial transaction data estimate the GHG emissions from what is bought by the user, the users must be given control over which purchases that ought to be included and offer a possibility to remove purchases that they feel should not be attributed to them. This is especially important for households with more than one adult who share costs between them. It is for example not uncommon that one member of the household is responsible for paying all bills, which complicates the GHG estimates. Allowing users to disregard some purchases or split them should solve a majority of such problems.

Third, in order to provide trustworthy results when using transaction data to calculate the GHG emissions of users, the workings of the system and the method for calculating emissions should be transparent and explained in the interface. Depending on the technology used, the explanation should probably also detail the certainty of the calculated results, and how specific/general the given emission estimate related to the purchases is.

Finally, in order to obtain long-term behavioural change, research has indicated that personal feedback (e.g., about people's carbon footprint) should be given frequently, rather than only once [39–42]. Indeed, the use of (continuous) transaction data are mainly beneficial to understand changes in behaviour that increase or decrease emissions. Since this requires users to use the calculator on a regular basis, mechanisms need to be in place to attract users to interact with the calculator over time. One such mechanism implemented in Svalna is the option to form groups with, for instance, friends or colleagues at work. The group members can see their combined emissions, compare themselves to each other, and set a common goal. By adding this social feature to the continuous consumption data, it allows a group of people to make use of the service as a tracker in the background, while focusing on actual changes. This social feature connects to several other social aspects of motivation and behavioural change such as norm activation, feelings of pride and guilt, peer pressure, and so on. While several studies with limited results have been conducted on carbon calculators in the past (e.g., [6,27–29]), this social setting is a rather unexplored area of carbon calculators (see [27]), which has some potential.

### 6.2. Limitations and Future Work

While our study has investigated users' attitudes towards Svalna, a major limitation is that we have not investigated whether Svalna causes users to reduce their emissions significantly. We aim to study this in the near future when we have gathered data from a larger group of users over a longer period of time.

Furthermore, as Salo et al. [6] have pointed out, a major problem with carbon calculators is the recruitment of users. Although Svalna currently has over 16,000 registered users, we sent out our survey only to the 2100 users who had been active the last three months and the percentage of users answering our survey was rather low. It is therefore not unlikely that our findings mainly pertain to the more active part of our user base. For example, while 70% of our respondents chose to connect their bank account, the percentage of all users connecting their bank account in Svalna is currently around 50%. For people who have not actively chosen to use a carbon footprint calculator, this percentage may be even lower. Svalna is therefore currently in the process of setting up further collaborations to spread the service to other potential users through campaigns or workplace competitions and collaborations. This development is interesting as it will allow us to analyse and understand reactions to Svalna from people who have not actively chosen to use the service.

Furthermore, although Svalna could potentially be rolled out in other countries, our results do not necessarily pertain to those countries. Sweden, where Svalna was started, is a country where more

than 60% of respondents in the most recent World Value Survey think that people can be trusted [43]. Further research is thus needed to determine how concerns about e.g., safety issues can be addressed in the best way, in Sweden and elsewhere.

Another limitation is the fact that Svalna experienced several technical problems with the connection to banks and a sometimes unreliable categorisation of transactions. Our results should therefore be seen as tentative and future research on more mature technologies is probably needed. However, many of the technical problems have currently been resolved. It would therefore be interesting to send out a similar survey to investigate the users' experiences.

Based on the results from this research project, we aim to look closer at ways to increase users' trust in systems using transaction data to calculate the carbon footprint of people, for instance through the introduction of more options to customise and personalise the calculations. We also aim to study the effects of carbon calculators such as Svalna through larger studies, taking into account people's underlying values and identities, as they have been shown to be good predictors of pro-environmental behaviour [44]. Additionally, such larger studies will also allow us to investigate whether the trust in the calculated GHG emissions changes through a mechanism using transaction data as used by Svalna. Additionally, this would allow us to determine if there are differences in trust between e.g., different age groups or related to gender.

Finally, Svalna is implementing different group features that may motivate users to interact with the app on a more regular basis. Continuous transaction data can play a role in supporting such group functionality as group members can see their progress. The effects of such functionality will be investigated in some follow-up studies.

**Author Contributions:** Conceptualization, A.B.-H. and D.A.; formal analysis, A.B.-H. and W.B.; investigation, A.B.-H.; resources, D.A.; data curation, A.B.-H.; Writing—Original draft preparation, W.B.; Writing—Review and editing, A.B.-H. and D.A.; visualization, A.B.-H.; supervision, W.B. and D.A.; funding acquisition, D.A. All authors have read and agreed to the published version of the manuscript.

**Funding:** This research is part of the programme Mistra Sustainable Consumption, funded by Mistra—The Swedish Foundation for Strategic Environmental Research. This research is also partly funded by the Irène Curie Fellowship granted to Wolmet Barendregt.

**Conflicts of Interest:** One of the authors is the founder of Svalna. Another author has worked on the development of Svalna. The first author has no formal relation to Svalna. The funders had no role in the design of the study; in the collection, analyses, or interpretation of data; in the writing of the manuscript, or in the decision to publish the results.

## Abbreviations

The following abbreviations are used in this manuscript:

| | |
|---|---|
| GHG | Greenhouse Gas |
| NDCs | Nationally Determined Contributions |
| M | Mean |
| SD | Standard Deviation |
| n | Number of Respondents |
| WWF | World Wildlife Fund |

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
