# Peer review of "Users’ Experiences with the Use of Transaction Data to Estimate Consumption-Based Emissions in a Carbon Calculator"

_sustainability, doi:10.3390/su12187777_

Round 1

Reviewer 1 Report

This is a preliminary study performed at the stage of the development of the calculator with a minor number of responding users. Unfortunately, I cannot see what is the relevance of this work. Authors, correctly, mention as limitation that results cannot be generalized, but the results do not have any validity for the specific (Anonymous) carbon calculator either. The research process itself does not in this form have any validity for research novelty. The added value of this exercise remains very low. I would propose the authors to use this as a preliminary process and use the received experience in a deeper study on the present stage of the (Anonymous) calculator and aiming to remarkably higher number of respondent in order to go deeper in investigating the different respondent groups. The aim of this approach would be the strategic design of responder/consumer oriented carbon calculator.

Author Response

Thank you for reviewing our manuscript. We are aware that this is a preliminary study. However, we have tried to clarify in the paper that the calculator we have investigated is a new type of calculator, with several actors currently aiming to develop similar kinds of solutions. Since not much research has been done on how users experience such a type of calculator, we think it is useful for such actors to learn from our experiences.

Reviewer 2 Report

The authors have submitted an interesting manuscript about users’ perception of a supposedly existing (but anonymous) carbon footprint calculator. The analysis is based on a short on-line survey.

I find strong points in the manuscript:

  • It covers an interesting topic with a down-top approach.
  • It is elegantly written.
  • The manuscript has been highly refined. No significant mistakes or errors.
  • The development of the paper is coherent with the scope that is defined at the beginning of the text.

With the proposed scope in mind, I have only a few minor remarks to the authors:

  • Please, specify always the meaning of symbols and acronyms before their first appearance (GHG, M, SD, N)
  • What is calculated: Carbon (title) or GHG? Please be more specific along the text.
  • The users are classified depending on personal data. However, no analysis of the results based on user’s differences is provided. I would be interesting to cover these aspects (e.g. how user’s age affects usage or trust)
  • A deeper review, commenting previous published works on similar carbon calculators would also benefit the paper.
  • “Conclusion and discussion”. Should be “Discussion and conclusion”? However, it would preferable to split in two: Discussion and Conclusions, being the latter a very short chapter in which the main results are highlighted and the limitations and future works are also indicated.

However, I personally think that at least two additional aspects should be discussed in the work:

  • How are carbon emissions calculated from consumption data? It is said that “a detailed account of how [Anonymous] determines emissions for each transaction within each category can be found in [Anonymous for review]” I cannot have any opinion or made verifications on this.
  • Are these results better than those obtained without consumption data? How much? Demonstration. Discussion. Conclusion.

Author Response

Thank you for the thorough review. We really feel the review has helped us to clarify some things and improve the paper. Below, we will address all comments one by one in italic text.

  1. Please, specify always the meaning of symbols and acronyms before their first appearance (GHG, M, SD, N). We have specified all acronyms in the text when they first appear. There is also a table towards the end of the manuscript with all abbreviations (L470).
  2. What is calculated: Carbon (title) or GHG? Please be more specific along the text. Although Svalna calculates GHG, the common term for this kind of tool is carbon calculator or carbon footprint calculator. We have added a new paragraph called 2.1 What is a Carbon Calculator? explaining the use of these terms and we have checked the whole document to make sure we consistently use GHG emissions when we refer to the outcomes of Svalna, while we keep the term carbon (footprint) calculator to refer to the type of tool we are studying.
  3. The users are classified depending on personal data. However, no analysis of the results based on user’s differences is provided. I would be interesting to cover these aspects (e.g. how user’s age affects usage or trust). While we agree that this would be interesting, we think our sample is too small to do so. However, we are now doing some more quantitative studies where we will both look at the effects as well as the variables mentioned here. We have therefore addressed this in the future work section (L451-453).
  4. A deeper review, commenting previous published works on similar carbon calculators would also benefit the paper. We have tried to cover the existing papers on carbon calculators in the background section. However, these papers cover different kinds of calculators, or only look at the workings of a number of calculators. To our knowledge, there are no scientific papers that look at carbon calculators that use continuous transaction data. We have tried to make it clearer what the difference between Svalna and the previously studied carbon calculators is to explain why the findings of previous studies are not wholly relevant to Svalna and why there is lack of studies on similar carbon calculators. This can be found in L119-127.
  5. “Conclusion and discussion”. Should be “Discussion and conclusion”? However, it would preferable to split in two: Discussion and Conclusions, being the latter a very short chapter in which the main results are highlighted and the limitations and future works are also indicated. We have split this section into two parts as suggested, one with a discussion (starting at L289) and one with a short conclusion as well as the limitations and future work (starting at L376).

However, I personally think that at least two additional aspects should be discussed in the work:

  • How are carbon emissions calculated from consumption data? It is said that “a detailed account of how [Anonymous] determines emissions for each transaction within each category can be found in [Anonymous for review]” I cannot have any opinion or made verifications on this. We have de-anonymised the draft and have indicated that the description of how all calculations are done in detail can be found in the paper we are referencing (L171).

  • Are these results better than those obtained without consumption data? How much? Demonstration. Discussion. Conclusion. We have added the following paragraph in L177-187 to discuss whether the results of a calculator like Svalna are better than those of a more traditional calculator: Svalna's solution largely relies on transaction data and registry data to estimate GHG emissions, but is this really more reliable? There is, as far as we know, no established standard regarding the calculation of personal carbon footprints (Wiedmann) which makes it hard to evaluate and compare different carbon calculators. However, Birnik has developed a set of assessment criteria for online carbon calculators. These criteria were used by Andersson to evaluate Svalna where he concluded that Svalna performs "over and beyond" these criteria. It seems that solutions using transaction data have the potential to capture a significantly larger share of users' spendings and hence GHG emissions in a more correct and controllable way. By relying on "external" data sources instead of the user's recollections this approach also avoids the risk of response biases that are likely to affect users who input data themselves. Until standards evolve and systematic comparisons between a set of carbon calculators is conducted this remains an open issue.

Reviewer 3 Report

1.        As the details of how the calculations were done was included in [Anonymous for review) but this was not available, it is not possible to comment on this aspect of the paper

2.       If the app was meant to be anonymous, then someone should have done a better job of removing the Svalna logo from the screenshots of the app

3.       L1-16 - It would be useful for the Abstract to mention that the calculator uses EE MRIO, and the research questions or aims (or both).

4.       L5332-63Carbon calculators – there are also Supermarket chains and Banks that have apps with more specific details

5.       Introduction – should include some information on EE-MRIO tables and their limitations eg under-estimation of emissions from meat from ruminants

6.       Introduction – could also mention somewhere that Sweden is a country where, like Denmark, there is generally a high level of trust, this might also have a bearing on the results

7.       Introduction – would be useful to mention some of the method outlined in L232 for changing behaviour eg moral nudge, norm activation, goal setting etc

8.       L93 – it would be useful to state explicitly how cash withdrawals and purchases were covered, as opposed to credit/debit card or mobilepay transactions?

9.       L97 – it would be useful to specify exactly what data was obtained from government agencies eg public transport data (transportation)? Residential energy consumption?

10.   L120-122 – 4 main categories – it is not clear how “1) consumption of good and services” is different to “4) food & beverages”. And for residential energy, how would EV charging be disaggregated and included in transport?

11.   L122-123 – as per comment 1, some detail on how the different types of transactions were handled could be included here, even if just in a diagram.

12.   L140 – participants – make it clear that the questionnaire was sent all the registered users at that time

13.   L140-150 – is it known whether the composition of the respondents is different to the registered users eg gender, age, employment status

14.   L153 – SPSS – please spell out this acronym

15.   L176  – mention that it is bank transaction data and official registry data from government agencies

16.   Figure 4 – it might be useful if you reversed the order it strongly agree on the far left – make it more obvious who agreed, as per the title, rather than who didn’t agree

17.   Do people need to check in regularly to reduce their impact, or log in, make the necessary changes and maintain them over time? Please provide some references for why you state that users need to continually check in to reduce their emissions (ie what aspect of maintaining behavioural changes this relates to)

18.   Further work – is linking data from supermarket chains included?

Author Response

Thank you for the thorough review of our manuscript. We have used the suggestions to significantly improve the paper. Below, we will answer all questions raised one by one in italics.

·      1. As the details of how the calculations were done was included in [Anonymous for review) but this was not available, it is not possible to comment on this aspect of the paper.
We have de-anonymised the reference to the paper in which the inner workings of the calculations in Svalna are explained in more detail (L171).

·      2. If the app was meant to be anonymous, then someone should have done a better job of removing the Svalna logo from the screenshots of the app.
Sorry, we should have caught that. The paper is now no longer anonymous so we hope this is not a problem anymore.

·      3.  L1-16 - It would be useful for the Abstract to mention that the calculator uses EE MRIO, and the research questions or aims (or both).
We have added the use of EE MRIO as well as the research questions to the abstract (L8-13). Compared to the previous version, we have changed the order of the last two research questions to make it more logical in the abstract and for consistency we have adapted this order throughout the whole paper.

·       4. L5332-63 Carbon calculators – there are also Supermarket chains and Banks that have apps with more specific details.
We have added this to the introduction (L62-70), indicating that we are aware of their existence. Indeed, they are a type of calculator that goes in a similar direction as Svalna. However, to our knowledge, no research has been done on those kinds of calculators, which is why we think our paper can be of interest to others.

·       5. Introduction – should include some information on EE-MRIO tables and their limitations eg under-estimation of emissions from meat from ruminants.
We are aware of the general limitations of using EE-MRIO data and have added a sentence on this in the discussion on L283-285. It reads like this: “EE-MRIO data implicitly assumes homogeneity with respect to price and product/service, while in reality the economy provides a variety of goods and services where the price is not always a good proxy for emissions.”

·      6.  Introduction – could also mention somewhere that Sweden is a country where, like Denmark, there is generally a high level of trust, this might also have a bearing on the results.
We have added this to the discussion section as a limitation, L436-440.

·      7. Introduction – would be useful to mention some of the method outlined in L232 for changing behaviour eg moral nudge, norm activation, goal setting etc.
We have briefly mentioned methods for promoting pro-environmental behaviour in people and their use in calculators in the introduction, L42-45.

·      8.  L93 – it would be useful to state explicitly how cash withdrawals and purchases were covered, as opposed to credit/debit card or mobilepay transactions?
Although we are aware that the reader may have many questions about the exact calculations done in Svalna, we are hesitant to add all details to this paper, as they were already described in Andersson (2020) and adding them could be considered a form of self-plagiarism. We hope de-anonymizing the paper helps the interested readers to find this information by themselves.

·       9. L97 – it would be useful to specify exactly what data was obtained from government agencies eg public transport data (transportation)? Residential energy consumption? We have separated the part about government agencies into two, making explicit which government agencies are meant and what data is retrieved through these agencies. We have also added what kind of data was received from the users themselves. In order to make the text more readable with these longer explanation, we have made it into a numbered list. This can be found in L134-144.

·      10. L120-122 – 4 main categories – it is not clear how “1) consumption of good and services” is different to “4) food & beverages”. We have indicated that the difference between 1 and 4 is the kind of store the purchase is made in, using food-store and non-food store categories. This can be found in L169-170 in parentheses.

And for residential energy, how would EV charging be disaggregated and included in transport? It is not possible to distinguish this, it will be visible in the increased energy consumption. We refer the reader to Andersson (2020) for all details about the calculation.

·      11.  L122-123 – as per comment 1, some detail on how the different types of transactions were handled could be included here, even if just in a diagram.
This is explained in detail by Andersson (2020) and we believe readers of the current manuscript who are interested in issues related to calculations are better helped by reading Andersson 2020, therefore we have made no changes in the manuscript.

·     12.  L140 – participants – make it clear that the questionnaire was sent all the registered users at that time. This is explained in more detail in the following way (L200-201): The questionnaire was sent to roughly 2100 active users of Svalna on the 11th of September 2019. Active users in this regard were defined as users who had logged in or created a new account during the last three months up until the survey.

·       13.   L140-150 – is it known whether the composition of the respondents is different to the registered users eg gender, age, employment status. We have added a comparison of our sample with the characteristics of the current user base to the paper using the following text in L211-224: The characteristics of our sample were similar to the group of all active users at the time, where 33% were women, 28% men, and 39% unknown. Concerning age, only the age of 55% of the users was known. The majority (22%) was between 25 and 34 years old, with smaller groups among the other age intervals (11% 18-24, 13% 35-44, 9% 45-54, 4% 55-64, 2% 65+)

·       14.   L153 – SPSS – please spell out this acronym. We have done that in line 217.

·       15.   L176  – mention that it is bank transaction data and official registry data from government agencies. We have changed this in L241-242.

·       16.   Figure 4 – it might be useful if you reversed the order it strongly agree on the far left – make it more obvious who agreed, as per the title, rather than who didn’t agree. Yes, that is a good idea. We have changed this accordingly (page 8).

·       17.   Do people need to check in regularly to reduce their impact, or log in, make the necessary changes and maintain them over time? Please provide some references for why you state that users need to continually check in to reduce their emissions (ie what aspect of maintaining behavioural changes this relates to).
We have added several references that indicate that continuous feedback is important for behaviour change (L408). We have also indicated how transaction data could be used to support e.g., group functionality (from L411).

·       18.   Further work – is linking data from supermarket chains included? This has been Svalna’s intention for a couple of years, but this requires collaboration with food retailers. While this is an option, we do not expect this to happen in the near future and have therefore not included it in the manuscript.

Round 2

Reviewer 1 Report

Unfortunately, I can not follow the logic of this paragraph: "Most carbon footprint calculators aim to estimate a user’s GHG emissions based on questions about e.g., traveling, heating and eating patterns (e.g., vegan, vegetarian, omnivore). However, consumption plays an important role in the size of an individual’s emissions, especially in wealthy countries [17]. Some carbon calculators, such as the calculator by Milieu Centraal [18], therefore explicitly state that their results do not take such emissions into account."

Author Response

Thank you for reviewing our paper again. We have clarified the logic of the paragraph you indicated as unclear. The following text is the revised paragraph:

Most carbon footprint calculators aim to estimate a user’s GHG emissions based on questions about e.g., traveling, heating and eating patterns (e.g., vegan, vegetarian, omnivore). However, consumption of goods, such as clothes and mobile phones, also plays an important role in the size of an individual’s emissions, especially in wealthy countries [17]. This is a factor that most carbon
footprint calculators do not take into account, and some carbon calculators, such as the calculator by Milieu Centraal [18], therefore explicitly acknowledge that their results do not include emissions based on the purchase of goods.

The revised paragraph is highlighted in the paper.

Kind regards

Reviewer 3 Report

Looks as though most of my comments have been included, thank you for clarifying those details, very interesting!

Author Response

Dear reviewer,

Thank you for taking the time to read our revised paper. Your comments were very helpful so we are happy that you think we have addressed them properly.

Kind regards